# Comparison of Outlier-Tolerant Models for Measuring Visual Complexity

**DOI:** 10.3390/e22040488

**Published:** 2020-04-24

**Authors:** Adrian Carballal, Carlos Fernandez-Lozano, Nereida Rodriguez-Fernandez, Iria Santos, Juan Romero

**Affiliations:** 1CITIC-Research Center of Information and Communication Technologies, University of A Coruña, 15071 A Coruña, Spain; carlos.fernandez@udc.es (C.F.-L.); nereida.rodriguezf@udc.es (N.R.-F.); Iria.santos@udc.es (I.S.); juan.romero1@udc.es (J.R.); 2Department of Computer Science and Information Technologies, Faculty of Computer Science, University of A Coruña, Campus Elviña s/n, 15071 A Coruña, Spain; 3Department of Computer Science and Information Technologies, Faculty of Communication Science, University of A Coruña, Campus Elviña s/n, 15071 A Coruña, Spain

**Keywords:** machine learning, sisual complexity, visual stimuli, correlation, human-computer interaction, compression error, psychiatry and psychology

## Abstract

Providing the visual complexity of an image in terms of impact or aesthetic preference can be of great applicability in areas such as psychology or marketing. To this end, certain areas such as Computer Vision have focused on identifying features and computational models that allow for satisfactory results. This paper studies the application of recent ML models using input images evaluated by humans and characterized by features related to visual complexity. According to the experiments carried out, it was confirmed that one of these methods, Correlation by Genetic Search (CGS), based on the search for minimum sets of features that maximize the correlation of the model with respect to the input data, predicted human ratings of image visual complexity better than any other model referenced to date in terms of correlation, RMSE or minimum number of features required by the model. In addition, the variability of these terms were studied eliminating images considered as outliers in previous studies, observing the robustness of the method when selecting the most important variables to make the prediction.

## 1. Introduction

Quantifying the visual complexity of an image is a task that is still very complex for the research community in Computer Vision. In a society full of visual stimuli, having the complexity value of an image could be a valuable contribution to work on visual impact, for example. Research groups from different areas of knowledge have focused on identifying features that are highly correlated with human aesthetic and visual preferences. Research has shown that different perceptual features, such as color, color combination, contour, or symmetry, influence people’s visual preferences and affective responses [1,2,3]. Complexity is considered to have a strong impact on preference and affection, given its relationship with excitement [4,5] and has therefore gained great relevance in psychological models of aesthetic appreciation [4,6]. Certain features of the human mind directly influence the sensation of perceived complexity [7]. Many factors, such as previous exposure to certain visual stimuli or familiarity with the given stimuli, affect the human perception of visual complexity. Authors such as Frijda (1989) or Blood & Zatorre (2001) also state that the aesthetic experience is affective or even emotional [8,9]. Therefore, there is no doubt that the perception of complexity is inevitably subjective, it can vary between different people and even with different circumstantial factors such as fatigue. Thus, the creation of precise computational models for the prediction of visual complexity is a difficult task to execute, since it may require the modeling of the visual cortex and even personal experiences. Leder et al. (2004) created a model that differentiates between aesthetic emotion and aesthetic judgments as two types of output. This model offers researchers the flexibility to consider and control for variables such as the level of knowledge and affective state of the recipients [10,11]. Graf and Landwehr [12] also contribute to the literature on visual preference with a model that allows aesthetic preferences to be formed by two processes that they consider to correspond to two levels of hierarchical processing. The first processing is stimulus-driven and reflects aesthetic evaluations of pleasure or displeasure. The second processing is conditionally activated and is characterized by being driven by the perceiver and may result in aesthetic evaluations of interest, boredom or confusion.

Berlyne [4,13] was the first scientist to provide an adequate psychobiological explanation of the effects of complexity on preference. Berlyne [4] analyzed the result of the interaction of reward and aversion system, and deduced that these systems would lead people to prefer intermediate levels of complexity, defined by characteristics such as the regularity of the pattern, the amount of elements, their heterogeneity or the irregularity of forms [4,13,14,15]. According to Berlyne, the lack of organization and the amount of elements is also considered a type of complexity. However, more recent research like Nadal’s [16] has shown that the results obtained by Berlyne were firmly conditioned by the experimental method chosen to measure complexity. Berlyne [4,13] argued that preference and interest increased linearly with visual complexity until an optimal level of excitation was reached. At this point, an increase in complexity would result in a decrease in excitation and preference would decrease. In other words, people seek to maintain a level of arousal coherent with their preferred level of arousal, any value above or below that level loses the person’s interest. Birkhoff [17] also published a large number of psychological articles proposing relationships between visual complexity and aesthetic value. Recently, other authors such as Van Geert and Wagemans [18] also support the importance of complexity and order in predicting aesthetic appreciation.

Forsythe (2011) based his work on the dataset created by Cela-Conde et al. [19]. They used this set to measure the correlation between the visual complexity perceived by the human being and the different measurements obtained by computer. Machado et al. (2015) [20] used the same dataset in their work, using artificial neural networks (ANN) and computer-generated metrics with Machine Learning to predict the visual complexity of images. In 2019, Fernandez-Lozano et al. [21] used the same dataset to assess the value of visual complexity through the application and validation of a set of Machine Learning models. They calculated more than 300 features based on the image compression error and Zipf’s law on three color channels. In this paper we rely on these three works mentioned to carry out a study of the area.

The remainder of the paper is organized as follows: Section 2 presents different state-of-the-art works; Section 3 describes the dataset and methods proposed; the results obtained in the experiments performed are shown in Section 4; Section 5 and Section 6 show the discussion and the conclusions respectively.

## 2. State of the Art

It has long been considered that two of the features that influence the perception of complexity—order and variety—determine beauty. For Tatarkiewicz [22], beauty also arises from “unity in variety”. Fechner [6] introduced the importance of two different forces—sometimes opposite—in experimental psychology and formulated the “principle of unitary connection of the variety” [23]. This principle argued that the stimuli were pleasant when they adequately balanced complexity and order. On the other hand, Birkhoff [17] was the first to formulate this relationship between order and complexity in mathematical terms. He defined order based on repetition and redundancy, and complexity as an expression of multiplicity. Thus, the complexity formulated by Birkhoff has the following form: M = O / C, where “M” is the average or the aesthetic value, “O” is the aesthetic order and “C” is the complexity. In other words, beauty increases when complexity decreases. Nevertheless, studies carried out by Eysenck [24,25] on the correlation between the aesthetic value resulting from the Birkhoff’s equation [17] and the assessment of a group of humans, suggested that both order and complexity contributed positively to the appreciation of beauty.

The computational measures mentioned have also been applied to artworks in an attempt to quantify their complexity. For example, Forsythe et al. [26] analyzed the correlation between the complexity value given by people to 800 visual stimuli and the results of JPEG and GIF compression measures and a Perimeter Detection measure. The JPEG and GIF compression measures are based on compress file lengths, so they are an estimate of the amount of information in the image. It is also possible to relate compression measures with Kolmogórov complexity, that measures ’the length of a shortest computer program (in a predetermined programming language) that produces the object as output’, since the compression method can be view as a way to get a compact description of the image. Their results showed that the three computational measures were significantly correlated with the complexity judged by people. The GIF compression showed the strongest correlation (rs = 0.74) and the Perimeter Detection showed the weakest (rs = 0.58), although certain differences were detected according to the type of stimulus assessed.

The first methods created to predict visual complexity were based on the number of elements and their regularity and heterogeneity [17,27]. The stimuli used in the first complexity prediction experiments were formed by different sets of polygons, which allowed to manually count the contained elements. In 2006, Lempel and Ziv [28] developed a new algorithm to measure visual complexity. This algorithm stands out for having been created from the smallest computer program required to store/produce an image. This method follows the theory that the minimum code length needed to describe an image is an appropriate measure of complexity [29].

Donderi [30] deduced that algorithmic information theory directly influenced the suitability of compression techniques to predict the complexity value. Some works such as Forsythe’s [26] and Marin’s [5] showed that methods such as Canny or Perimeter Detection, based on phase congruence, are also optimal for measuring visual complexity. Other studies on experimental aesthetics published recently have focused on investigating how the exposure time of the stimulus affects image processing [31]. To do this, the intrinsic and extrinsic factors that affect the human response on the aesthetic value of the images were examined [32], as well as the way in which the “good composition” or the “visual rightness” is revealed according with edge orientation and luminance [33].

The most commonly used method to determine the perceived complexity value is image evaluation by a group of humans [34,35]. Following this methodological line, Forsythe et al. [26] conducted an experiment to analyze the performance of a set of metrics related to GIF and JPEG 2000 compression and to Perimeter Detection. A total of 240 humans participated in the study and evaluated a set of more than 800 images from their perception of visual complexity within a given score range. [19]. According to the authors, GIF compression offered the best correlation results with the evaluation carried out by humans, reaching an R-Squared = 0.5476. In the field of aesthetic psychology several works were also developed in which different images were evaluated by humans according to exclusively aesthetic criteria [26,36,37]. In 2016, Schettino [38] recruited seventeen men (between 19 and 33 years old) to evaluate one hundred images representing different everyday scenes (for example, people in the supermarket, in the restaurant, playing music or exercising), as well as naked female bodies and heterosexual interactions, selected from IAPS [39]. In the study conducted by Street et al. [40], 228 men and 204 women, aged 17 to 88, participated, evaluating 81 abstract monochrome fractal images (9 complete sets of 9 FD interactions) generated by the midpoint shift technique. In addition, Lyssenko et al. [41] conducted a study with 19 participants aged 19 to 37 years, who evaluated a set of 79 images belonging to a collection of 150 gathered by Hayn-Leichsenring et al. [42]. Friedenberg and Liby [43], meanwhile, selected a group consisting of 5 men and 20 women students from Manhattan College in New York to evaluate 10 images.

In this context, Marin y Leder [5] turned to the International Affective Picture System to select a set of stimuli with which they studied different computational measures related to the complicity classifications of different materials. With this, they found that TIFF compression (R-Squared = 0.2809) and JPEG compression (R-Squared = 0.2704) correlated more than Perimeter Detection with values of subjective complexity.

Nadal et al. [44] introduced a new conception of aesthetic sensitivity: the degree to which someone’s aesthetic assessment is influenced by a given feature. In the same research, Nadal et al. [44], found that Visual Aesthetic Sensitivity (VAST) presented several psychometric problems, so they performed two different experiments; in experiment 1 they characterized aesthetic sensitivity with complexity, symmetry, contour and balance; in experiment 2 they replicated the findings of experiment 1 and evaluated the test-retest reliability of an instrument designed to measure aesthetic sensitivity. From the results of this study it can be deduced that people differ significantly in the extent to which visual features influence their taste.

In 2015, Machado et al. [20] proposed new complexity estimates based on the compression error and the Zipf’s law [45]. Different edge detection filters were applied in all color channels to calculate the features in each image. To do this, they previously examined the performance of the Sobel [46] and Canny [47]. As a result of using the filters mentioned above, a total of 329 features were extracted based on seven metrics that were applied to the three color channels [48]. The estimates reached by Machado et al. [20] bear some similarity to the Perimeter Detection method used by Forsythe et al. [26]. To reach these estimates, the percentage of pixels corresponding to the edges in the image was measured. This method reported better performance that the state-of-the-art. On the other hand, metrics similar to those obtained by Forsythe et al. [26] also provided very similar results using GIF compression based on JPEG and Fractal compression error without the application of edge detection. The features related to edge quantification gave better results than the Perimeter Detection method used by Forsythe et al. [26] in estimating visual complexity. Finally, JPEG compression with previous application of Canny (R-Squared = 0.5868) and Sobel (R-Squared = 0.5944) edge detection filters obtained the highest correlations. The objective of Machado et al. [20] was to improve the results in prediction of perceived complexity value by humans by using a computational method. According to their work, edge density and compression error offer the best results in the task of predicting visual complexity. With this, they suggest that the psychological processes that affect edge detection and information processing play a very important role in the subjective perception of complexity. In addition, they tried to assess the correlation results of individual perception by combining their vector features with artificial neural networks (ANN). As a result, they obtained an increase of R-Squared to 0.6939 using all the features and inputs proposed in an MLP. The use of 329 features together improved the correlation with a single metric. The authors concluded that the system that could use all the features worked better than any other input combination. Despite considering the work of Machado et al. [20] as a starting point for this work, the use of an ANN-based model is not always the best choice for prediction-related tasks. Neural Networks and, particularly, Multilayer Perceptrons, have to be over-adjusted, especially if a large number of samples are not available [49]. There are other methods that may be more suitable for prediction tasks and that minimize the problems discussed, such as combining predictors from many different models [50], such as those used by Fernandez-Lozano et al. [21], or “the drop-out technique”.

Fernandez-Lozano et al. [21] studied other computational methods in order to identify alternatives to ANNs already used to solve problems related to visual complexity. Until then, the best results obtained had been R-Squared = 0.69 using as an input set 329 features with a multilayer ANN [20]. Among the methods studied, Fernandez-Lozano et al. [21] concluded that the most reliable for the problem raised was FSMKL, which obtained a result of 0.71, emphasizing that such results could be obtained with only 22 metrics. These were repeated 50 times in independent experiments, obtaining an RMSE with little variability.

This study proposes other models based on Machine Learning that apply to the same input data used by Forsythe et al. [26], Machado et al. [20] and Fernandez-Lozano et al. [21].

## 3. Materials and Methods

The authors tested several different computational models following a robust experimental design for regression analysis [51] to evaluate the performance across different experiments of multiple machine learning computational models and to compare the statistical differences of the results. The performance of the models was evaluated using R-squared. Fifty runs of a 10-fold cross-validation experiment were performed. The purpose of cross-validation is to test the ability of the model to predict new data not used in its training so that issues like overfitting or selection bias can be detected as well as to provide an insight into how the model generalizes into a separate dataset. The results obtained were evaluated by analyzing the complete set of variables and eliminating those images that are outliers to validate the pre-processing of the dataset.

### 3.1. Stimuli and Participants

The dataset used was provided in the study carried out by Forsythe et al. [26]. The initial dataset came from Cela-Conde et al. [19] and contained about 1500 digitized images. The final standardized set had a total of 800 images that were divided into 5 different categories: artistic representative (RA), artistic abstract (AA), representational non-artistic (RN), abstract non-artistic (AN) and photographs of natural and human-made scenes (NHS). Examples of each category are presented in Figure 1. In the work carried out by Forsythe et al. [26], 240 people participated, among which 46.67% are men and 53.33% women, form the University of the Balearic Islands, without no formal artistic knowledge. Although this dataset is not of a public nature, it was provided to us by its original authors.

Each stimuli was set to 150 ppi, its size to 9 by 12 cm, the color scale in each of these images was changed to minimize the psychophysical variables and the stimulation’s luminance changed to range from 370 to 390 lx. For certain cases, for optimal anonymization, the author’s signature has been deleted manually. The 800 images were grouped into 8 sets of 100 pictures, each evenly distributed. Each participant was given a very low-complicated stimuli at the start of the experiment (an icon) and another with great complexity (a town) as examples [19].

In conjunction with Snodgrass and Vanderwart [52], participants were given a description of complexity as “the amount of detail or intricacy”. Then there was a digital monitor between 2 and 7 m from which 100 mages with a ratio of 16:9 and 400×225 cm were seen. Every 5 s one image was shown. The images were rated on a scale of 1 to 5 by each of the participants, 5 of them were very complex and 1 were very simple.

The dataset has been used previously in other research works and the presence of outliers [21] is known. The detection of the cases that assume extreme cases and that affect by modifying the regression coefficients of the analysis models was based mainly on three criteria (Cook’s distance, studentized residuals and high leverage points). In order to validate the robustness of the proposal, several hypothesis-contrast tests with adjustment for multiple comparisons were performed, resulting in the existence of different images (6) with a great capacity to modify the adjustment of the models and which, therefore, were suppressed in subsequent analyses.

### 3.2. Computational Models

The authors performed several experiments in order to select the best model using R [53] and the RRegrs package [54], MATLAB^®^ and Java-based CGS [55]. Some of the used computational models looked for the smallest subset of variables of the original set which provided a better performance, or at least equal to that obtained when using all the possible variables [56].

More specifically, the used methods were as follows: Elastic Net (ENET) [57], Feature Selection Multiple Kernel Learning (FSMKL) [21,58,59,60], Partial Least Squares Regression (PLS) [61], Random Forest (RF) [62], Random Forest–Recursive Feature Elimination (RF-RFE) [62,63], Support Vector Machines using radial functions (SVM-Radial) [64] and a recently proposed hybrid genetic algorithm based on regression models named Correlation by Genetic Search (CGS) [55]. A brief description of the algorithms is given below.

ENET regression combines the power of a Ridge and Lasso regression in a single algorithm. By means of a hyperparameter it is able to control which of the variables under analysis are relevant eliminating non-relevant sets (acting as Lasso) or even trying to reduce them to a number as close as possible to zero (acting as Ridge). It is a special type of regression algorithm that belongs to the group of regularization algorithms. It initially emerged as a critique of the Lasso model and its ability to select variables, which was sometimes very dependent on the data analyzed and, therefore, a high variability depending on the examples used from the dataset in the analysis.

FSMKL is a model based on kernels, widely used to solve different problems in very different fields. They usually perform well in high-dimensionality or noisy environments. In this work, the different subgroups of variables extracted from each individual image were coded in kernels and, finally, all the kernels were combined to solve a regression problem using a SVM. In addition, it performed an internal feature selection process on each subset by choosing the most relevant variables/subsets to solve the regression problem.

PLS is a widely used model when analyzing high-dimensionality and highly correlated data. However, the model is not very explanatory regarding the variables employed or with what degree of importance, being used mainly when the final result is the only relevant part of the analysis. The idea of its operation consists of reducing the number of variables as much as possible, selecting those variables with the least correlation between them and performing a least squares regression only on the reduced subset. To this end, the model uses linear combinations of the variables calculated with a principal component analysis (PCA) by selecting those projections that best describe the covariance of the data and the dependent variable. Therefore, PLS bases its operation on identifying a linear regression by projecting the prediction variables and the observable variables to a new space.

RF is an ensemble technique able to solve regression and classification problems by combining simple decision trees using an approach known as bagging. This means that RF will train each decision tree over a different set of data that is chosen for replacement. Therefore, the final result shown by the algorithm is a combination of simple decision trees, rather than the result given by only one of them. This allows us to understand the importance that each variable has in the final decision depending on the decision trees that have previously used it.

RF-RFE is a combination of the previously commented RF and to which an external feature selection phase known as Recursive Feature Elimination (RFE) is added. This basically refers to extracting in each iteration of the RF algorithm the importance of the variables in the final prediction and eliminating recursively those that have had less weight in the process, looking for the subset of variables that obtains the best result or, at least, the same result as the model that uses all the variables.

SVM-Radial is a Support Vector Machines (SVM) model for regression problems with a Radial Basis Function (RBF) kernel that attempts to identify the hyperplane which maximizes the separation margin; in the case of regression problems, it is necessary to add an error tolerance factor, since the outputs are real numbers. SVM bases its operation on the construction of an optimal hyperplane in the form of a decision surface, so that the margin of error is minimal in the estimation of the best regression. Some of the examples are used to calculate the location of the decision surface so that the least overall error is made, they are the support of the decision surface. The power of SVMs is mainly found in the kernel trick, which allows the user to increase the dimensionality of the data and thus transform the input space into a larger dimensional space without increasing the computational cost of the analysis.

CGS is a hybrid genetic algorithm based on regression models. Pérez et al. [65] used this algorithm developed by our research group to predict vertical urban growth in Tokyo, based on macro-economic data. Carballal et al. [55] also used CGS through transfer learning to predict aesthetics. An adjusted linear regression model is created with this method from the selection of features (Kudo et al. [66]), the transformation of features (Lui textit et al. Cite Liu: 1998: FTS: 630300.630355) and the parameter selection (Hurvich textit et al. cite Hurvich1990) applied simultaneously. This linear regression model uses any measure of performance correlation in the evolutionary process. The algorithm must determine the most important metrics. In addition, it must create a model established in a multiple regression that achieves the strongest relationship between the desired result and the result obtained in terms of correlation (Spearman, Person or Kendall), modifying the search space with the values selected during the process of filtration. The best combinations of transformations in the input variables that maximize the previously determined objective function are determined through the evolutionary process. The termination criteria of the evolutionary process are connected to the performance of the current individuals of a population in relation to the average individual of the population. When the average difference is lower than the pre-set threshold value for a homogeneous population, the iterative search process comes to an end (see Figure 2). The genetic algorithm uses mathematical operators to refactor input variables and thus find a suitable solution. Any continuous mathematical function could be used, but the selected subset was constructed by grid search, just like the other parameters used in this work.

## 4. Results

The results obtained during the experimental phase are shown in Figure 3 and correspond to 50 runs of a 10-fold cross-validation experiment. Results can be observed for the two datasets analyzed, the one that has the complete set of data and the one that removed the images identified as outliers. As shown in Figure 3, all models had better or at least more stable results when analyzing the dataset from which the outliers were removed, with the exception of ENET.

The improvement in the FSMKL model was also significant, being able to increase its performance according to R-squared by several percentage points. It can also be observed how CGS, the winning model, was the one that presented the most unstable behavior along the 50 repetitions but it was capable of surpassing all the others in a statistically significant way. This model offered greater variability in both cases (with and without outliers). Particularly relevant is the fact that one of the experiments was clearly deviated although it still offered a higher value than the average of the rest of the methods.

According to what is indicated in Reference [51], for the design of an experiment to be robust in general and, in particular, with the package RRegrs [54] it is necessary to compare the results obtained by means of a null hypothesis test. In this case, only the four best models of Figure 3 (RF, RF-RFE, SVM-Radial and CGS) were used and separated among problems (full dataset and dataset without outliers) as shown in Figure 4a,b. The Kruskal-Wallis paired test established, with a significance level of α=0.05, and assuming the null hypothesis that our results are statistically equal, that the null hypothesis was rejected with a *p*-value $<2.2\times 10^{-16}$. Therefore, the CGS model achieved the best performance score among the 50 different runs performed for each of the datasets used.

## 5. Discussion

Any of the new methods detailed in this paper offers better results than those seen in [21]. According to the results obtained, the CGS method is statistically better than any of the other methods studied. Considering the number of features used in the 50 independent runs, the average of necessary features is 23.08±0.85 in the case of using the complete set of images, and 23.21±0.89 if we eliminate the outliers detected in [21]. Remember that FSMKL had obtained an average R-Squared value of 0.71 when using 21.60 features. If we compare it with the rest of the proposed methods, the next one would be RF-RFE that would need 153.60±60.64 in the first case and 148.48±52.42 in the second.

As for the selected features, a total of 11 appeared in at least 50% of the 10-fold divisions of the 50 independent experiments, detailed in Table 1 following the nomenclature model used in [20].

As in References [20,21], the metrics related to the JPEG compression error were the most commonly selected by the method. Unlike the FSMKL method, where the use of the Canny edge detection filter and the Saturation color channel seemed to have more relevance, with the CGS method both that particular color channel and the use or not of edge detection filters were not so much a priority. FSMKL was able to use the small differences between the metrics (which were highly correlated) to obtain correlations of the order of 0.71. In the case of CGS, the method seemed to find better approximations using groups of features with a lower level of internal correlation compared to FSMKL.

As for the RMSE, the average of GCS in the experiments is 0.3944±0.1086 with outliers and 0.3949±0.1100 without outliers, improving with the next method (SVM-Radial) with 0.5140±0.0008 in the first case and 0.5137±0.0008 in the second. Like the R-Squared values, where the SVM-Radial variability was minimal, CGS was also statistically better in terms of RMSE although this is a method with greater variability.

## 6. Conclusions

This paper focused on the study of the suitability of different methods based on Machine Learning in a regression problem related to visual aesthetics. It was found that the CGS method (hybrid method for the creation of multiple regression models based on the maximization of the correlation) offered the best correlation results in terms of R-Squared, which were statistically corroborated through a Kruskal-Wallis paired test.

This method was also studied by eliminating from the set of study images already identified as outliers in previous studies, as well as the rest of the proposed methods. The results show that the method remains stable in statistical terms and the metrics that the method needs to obtain these results hardly vary. These evidences show the capacity of the model to identify metrics that are not dependent on possible outliers in the input set.

The choice of the CGS method, not only based on the results obtained, is appropriate since the input data have a relatively small dimension (329 features), which allows using equally small populations. This fact guarantees that this method does not fall in local minimums and the computational cost is reasonable (for more information [55]).

## Figures and Tables

**Figure 1 entropy-22-00488-f001:**
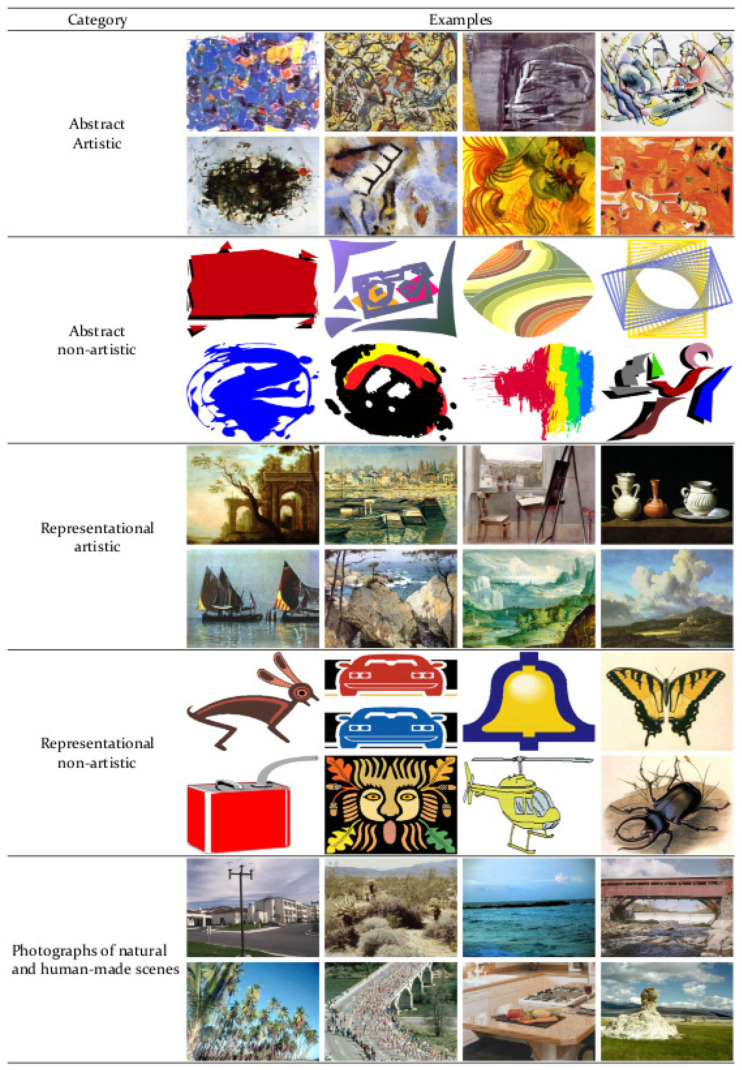
Examples of stimuli of each category.

**Figure 2 entropy-22-00488-f002:**
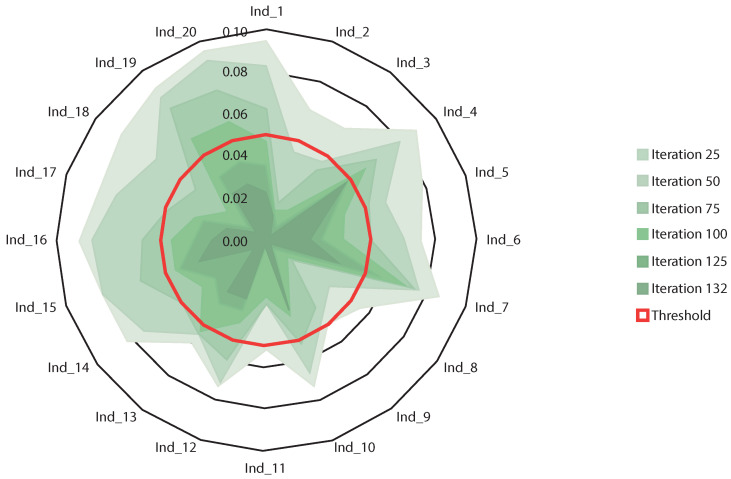
Example of the end of the evolutionary process. The example consists of 20 individuals and an average threshold of 0.05 (red line). Each individual is mentioned in the graphic as Ind_*X* (where *X* is the number of the individual). The shade of green is darker with more iterations. At the moment in which the average difference is less than the pre-set threshold value (0.05 in this case) the iterations end, together with the evolutionary process.

**Figure 3 entropy-22-00488-f003:**
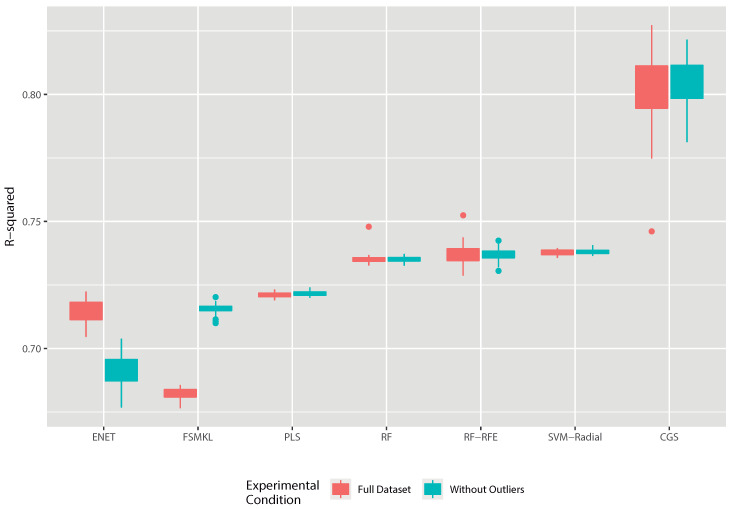
Results obtained in the experiments carried out with different ML models (Elastic Net (ENET), Feature Selection Multiple Kernel Learning (FSMKL), Partial Least Squares Regression (PLS), Random Forest (RF), Random Forest–Recursive Feature Elimination (RF-RFE), Support Vector Machine–Radial (SVM-Radial) and Correlation by Genetic Search (CGS). The behavior of each model is shown by analyzing two different datasets: dataset processed without the outliers and with the complete unprocessed set.

**Figure 4 entropy-22-00488-f004:**
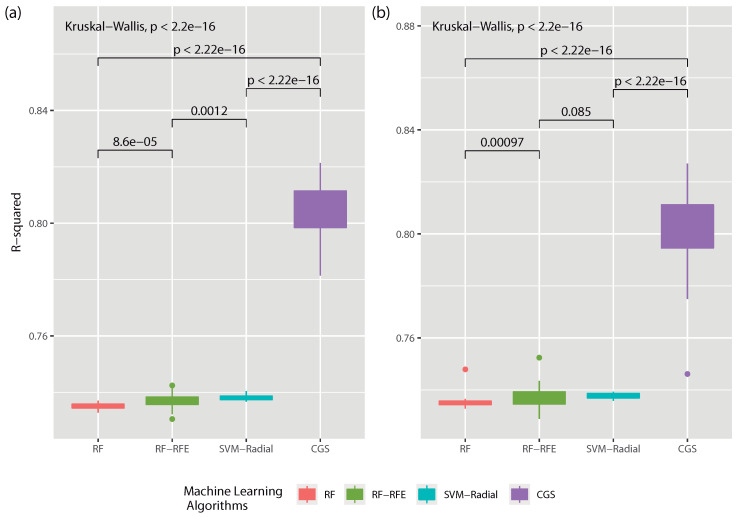
Statistical comparison among the 4 ML models (RF, RF-RFE, SVM-Radial and CGS) that obtained the best results during the analysis phase. You can see in (**a**) the comparison of the results once the outliers of the dataset are eliminated and in (**b**) the statistical comparison with the complete dataset, without any processing.

**Table 1 entropy-22-00488-t001:** Set of features used by CGS, organized by appearances. The identified features were: the average appearances in 10-fold CV 50 independent runs using all imagenes (With Outliers) and previously eliminating identified outliers (Without Outliers). All the features were identified using the terminology proposed in [20].

Feature	Avg Appearances	Avg Appearances
	With Outliers	Without Outliers
Rank(NoFilter(V), M)	100.00%	100.00%
Size(NoFilter(H), M)	99.60%	98.80%
JPEG(Canny(S), Low)	81.20%	81.80%
Size(NoFilter(H+CS), M)	80.20%	81.00%
STD(Canny(S), value)	80.20%	81.20%
Fractal(NoFilter(H), High)	79.60%	81.00%
JPEG(NoFilter(H), High)	66.20%	66.40%
Box-Counting(Canny(V), M)	62.00%	63.40%
JPEG(Canny(S), Medium)	61.00%	58.60%
JPEG(Canny(S), High)	58.60%	58.00%
Size(Canny(H+CS), M)	56.20%	58.00%

a Features previously identified by Machado et al. [20] as the best individual features for solving the problem. b Features previously identified by Fernandez-Lozano et al. [21] as the best individual features for solving the problem.

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
