# Peer review of "Comparison of Outlier-Tolerant Models for Measuring Visual Complexity"

_entropy, 2020, doi:10.3390/e22040488_

Round 1

Reviewer 1 Report

This is an interesting study that gives an important comparison and overview on machine learning methods predicting visual complexity. However, I think there are a few points mostly concerning the presentation that could be improved: 22-23: I think the references to psychological models of aesthetic appreciations should also include more recent models like Leder et al. (2004; doi:10.1348/0007126042369811), Leder and Nadal (2014, doi:10.1111/bjop.12084), Jacobsen (2006; doi:10.1162/leon.2006.39.2.155) and maybe also Graf and Landwehr (1015, doi:10.1177/1088868315574978) to name just a few. (Of course, the authors can decide which papers they want to cite. However, the two citations from 1876 and 1971 are in principle fine, but certainly do not reflect the current state of research.) Also Van Geert and Wagemans (2019; doi:10.1037/aca0000224) have very recently published a nice overview on order and complexity. 74: "ant" -> "and" 106: "y" -> "and" - As far as I understood, the identification of images as outliers has been done in other publications. However, it would increase understanding if at least a rough description how outliers have been identified would also be included in this paper. 174: As p values are notoriously hard to compare, I think it does not make much sense to speak of a "very significant value of p". - In general, the line width of the figures seems to be too small. Especially in Figure 2, where it is very hard to distinguish the colors. It might also help, if you would use a grey background as in Figure 1 and color filled box plots. - Table 1 seems to miss the horizontal lines. - Please number the four sub-figures in Figure 3 and explain in the caption that they represent different stages/iterations of convergence (if this is what they do represent).

Reviewer 2 Report

Unfortunately, as a social psychologist, this manuscript ended up being largely outside of my expertise, which includes collecting data from humans and doing statistical analysis on that data. I have done a lot of collaborative research with computer vision researchers, but I do not have expertise related to computational models.

Nonetheless, I will do my best to provide useful information to the authors and editor(s).

  1. Overall, I thought this was an interesting and informative paper, although as a non-expert my understanding of the contribution is weak. That is, I do not know if it makes a substantial contribution to scientific knowledge.
  2. I though the title should include something about the topic of aesthetic value.
  3. As a native American English speaker, I thought that the language needed substantial improvement. For example, what does “be a great contribution to work the visual impact” mean on line 17?
  4. On line 174, the paper says “rejected with a very significant p – value …”. However, statistical significance is a binary issue. If you reject the null hypothesis, the finding is “statistically significant.” If you fail to reject the null hypothesis, the result is “nonsignificant.” Moreover, the text goes on to claim that one model was “clearly the best”, but statistical significance only tells you where differences lie, not in how much better one is than another (that is instead an issue called “effect size”). For example, with a huge sample size, even a tiny effect can have a very small p value. The “clearly better” claim can be justified, but not based on a p value.
  5. Overall, the results seem to support the authors’ claims that CGS best predicted human aesthetic ratings in this image dataset. I think the generalizability of these findings would be strengthened if the authors could find another image dataset (completely separate images from the current set) that has human ratings of image aesthetics and did the same kinds of computational modeling and results analyses on them. That is, perhaps there was something idiosyncratic about this dataset that made the CGS excel over the alternative computational models (or perhaps such generalization is not a concern in this field of study).

Reviewer 3 Report

This paper aims to compare outlier-tolerant models for measuring visual complexity.
The authors described the literature and presented the correlation of some learning models for measuring visual complexity in images. I read the paper two times, and felt it will be hard for readers to understand in its current form. Both the paper structure and writing is not clear nor appropriate. 
I may offer the following comments:
Content:
1. very long title, stick to 10 words only and I recommend:
Comparison of outlier-tolerant models for measuring visual complexity.
2. No sample images provided?
3. The contribution is not clear, partially due to writing issue. For example the methods tested in Fig. 1, show that PLS method is also tolerant, but is excluded in Fig 2, why? 
4. What effect of outlier? more details on experiments should be added?
5. What is JPEG/ GIF compression measures? there also need to describe the basics of tested method, eg. PLS and SVM.
6. How did they measure the user perception, is that a public dataset data?

Style:
1. Material and methods section should be placed before results section and it should define all tested models briefly to the reader.
2. Figure 3 is not legible, i can not see the details?
3. No overview showing the paper sections and content at the end of introduction section.

Round 2

Reviewer 2 Report

2nd round review

I thought the paper is much improved. However, with clearer writing often other concerns are easier to detect. I think that is the case here. My concerns are such that I think they might not be big issues, or they might, I cannot say--because 2 of 3 are asking for information that was not provided in the prior drafts.

  1. Three main issues
    1. The paper starts off in a confusing way. The first sentence of the abstract and body suggests this research is about aesthetics, but elsewhere you discover it is really about complexity, like the title says. This is confusing. The abstract doesn’t mention complexity until about half way through, but complexity should be right up front in both the abstract and introduction.
    2. What is the outcome variable? Is it “visual complexity perceived by the human being” as suggested on line 55? If so, say in this paper how visual complexity was measured back in the original paper. In your “Stimuli and participants”, put in the question(s) and response scale(s) so the reader can figure out what your models are explaining without having to go back to the original research. That is, make your paper a stand-alone work that is easy to read. Likewise, the abstract should say what the outcome variable is. This could be done simply. For example, you could change “Correlation by Genetic Search (CGS) … obtained statistically better results than any other model referenced to date in terms of correlation…” to be “Correlation by Genetic Search (CGS) … predicted human ratings of image visual complexity better than any other model referenced to date in terms of correlation…”
    3. In your analyses, what are the variables and units of analysis? My guess is that you are correlating the mean human rating for each image with the predicted value for each image based on the computational model. That is: "Model predicted value for one image <predicts> mean human rating for that image". In that case, what is the sample size for your reported results? 50 runs x 800 images = 40,000 datapoints per analysis? Or are you doing something else, such as treating the resulting correlation for each run as the DV in a “meta  analysis”, and reporting the results of each study as a datapoint (e.g., in Figure 3)? That is, please clearly say for your reported results what are the units of analysis, and it might be clearest if you do so by explaining how the sample sizes are calculated (e.g., For the results in Fig. 3, the sample sizes for each box is 50, one per run). Likewise, what is the DV in Fig 4 (and its sample size)?

Reviewer 3 Report

The authors responded to my comments and the paper clarity improved significantly. 

Author Response

The reviewer has requested further comments.

We appreciate the comments made by this reviewer, as it helped to significantly improve the work presented.

Round 3

Reviewer 2 Report

Thank you for addressing my concerns. I think the paper is much easier to read and understand now.